# Anticipating and Adapting to the Future Impacts of Climate Change on the Health, Security and Welfare of Low Elevation Coastal Zone (LECZ) Communities in Southeastern USA

Thomas Allen [1], Joshua Behr [2], Anamaria Bukvic [3], Ryan S. D. Calder [4], Kiki Caruson [5], Charles Connor [6], Christopher D'Elia [7], David Dismukes [7], Robin Ersing [8], Rima Franklin [9], Jesse Goldstein [10], Jonathon Goodall [11], Scott Hemmerling [12], Jennifer Irish [13], Steven Lazarus [14], Derek Loftis [15], Mark Luther [16], Leigh McCallister [9], Karen McGlathery [17], Molly Mitchell [15], William Moore [18], Charles Reid Nichols [19], Karinna Nunez [15], Matthew Reidenbach [17], Julie Shortridge [20], Robert Weisberg [16], Robert Weiss [13], Lynn Donelson Wright [21,*,†], Meng Xia [22], Kehui Xu [7], Donald Young [9], Gary Zarillo [14] and Julie C. Zinnert [9]

[1] Department of Political Science and Geography, Old Dominion University, Norfolk, VA 23529, USA; tallen@odu.edu

[2] Virginia Modelling Analysis and Simulation Center, Old Dominion University, Norfolk, VA 23529, USA; jbehr@odu.edu

[3] Department of Geogaphy, Virginia Tech, Blacksburg, VA 24061, USA; ana.bukvic@vt.edu

[4] Department of Population Health Sciences, Virginia Tech, Blacksburg, VA 24061, USA; rsdc@vt.edu

[5] School of Interdisciplinary Global Studies, University of South Florida, Tampa, FL 33620, USA; kcaruson@usf.edu

[6] School of Geosciences, University of South Florida, Tampa, FL 33620, USA; cbconnor@usf.edu

[7] College of the Coast and Environment, Louisiana State University, Baton Rouge, LA 70803, USA; cdelia@lsu.edu (C.D.); dismukes@lsu.edu (D.D.); kxu@lsu.edu (K.X.)

[8] School of Public Affairs, University of South Florida, Tampa, FL 33620, USA; rersing@usf.edu

[9] Department of Biology, Virginia Commonwealth University, Richmond, VA 23284, USA; rbfranklin@vcu.edu (R.F.); slmccalliste@vcu.edu (L.M.); dyoung@vcu.edu (D.Y.); jczinnert@vcu.edu (J.C.Z.)

[10] Department of Sociology, Virginia Commonwealth University, Richmond, VA 23284, USA; jgoldstein2@vcu.edu

[11] Department of Engineering Systems and Environment, University of Virginia, Charlottesville, VA 22904, USA; goodall@virginia.edu

[12] The Water Institute of the Gulf, Baton Rouge, LA 70803, USA; shemmerling@thewaterinstitute.org

[13] Center for Coastal Studies and Civil and Environmental Engineering, Virginia Tech, Blacksburg, VA 24061, USA; jirish@vt.edu (J.I.); weiszr@vt.edu (R.W.)

[14] Department of Ocean Engineering and Marine Science, Florida Institute of Technology, Melbourne, FL 32901, USA; slazarus@fit.edu (S.L.); zarillo@fit.edu (G.Z.)

[15] Center for Coastal Resource Management, Virginia Institute of Marine Science, Gloucester Point, VA 23062, USA; jdloftis@vims.edu (D.L.); molly@vims.edu (M.M.); karinna@vims.edu (K.N.)

[16] College of Marine Science, University of South Florida, St. Petersburg, FL 33701, USA; mluther@usf.edu (M.L.); weisberg@usf.edu (R.W.)

[17] Department of Environmental Sciences, USA University of Virginia, Charlottesville, VA 22904, USA; kjm4k@virginia.edu (K.M.); reidenbach@virginia.edu (M.R.)

[18] Department of Atmospheric and Planetary Sciences, Hampton University, Hampton, VA 23668, USA; william.moore@hamptonu.edu

[19] Marine Information Resources Corporation, Queenstown, MD 21658, USA; rnichols@mirc-us.com

[20] Department of Biological Systems Engineering, Virginia Tech, Blacksburg, VA 24061, USA; jshortridge@vt.edu

[21] Southeastern Universities Research Association, Washington, DC 20005, USA

[22] Department of Natural Sciences, University of Maryland-Eastern Shore, Princess Anne, MD 21853, USA; mxia@umes.edu

\* Correspondence: ldwright@bellsouth.net
† Retired.

**Abstract:** Low elevation coastal zones (LECZ) are extensive throughout the southeastern United States. LECZ communities are threatened by inundation from sea level rise, storm surge, wetland degradation, land subsidence, and hydrological flooding. Communication among scientists, stakeholders, policy makers and minority and poor residents must improve. We must predict processes

spanning the ecological, physical, social, and health sciences. Communities need to address linkages of (1) human and socioeconomic vulnerabilities; (2) public health and safety; (3) economic concerns; (4) land loss; (5) wetland threats; and (6) coastal inundation. Essential capabilities must include a network to assemble and distribute data and model code to assess risk and its causes, support adaptive management, and improve the resiliency of communities. Better communication of information and understanding among residents and officials is essential. Here we review recent background literature on these matters and offer recommendations for integrating natural and social sciences. We advocate for a cyber-network of scientists, modelers, engineers, educators, and stakeholders from academia, federal state and local agencies, non-governmental organizations, residents, and the private sector. Our vision is to enhance future resilience of LECZ communities by offering approaches to mitigate hazards to human health, safety and welfare and reduce impacts to coastal residents and industries.

**Keywords:** sea level rise; land loss; low elevation coastal zones; social sciences; human health; adaptive management; cyber network; coastal resilience; computer modeling

## 1. Introduction

Coastal communities in the low-lying southeastern USA face increasing hazards, including compound flooding by coinciding or sequential storm surge and torrential rains and lingering post-event impacts on human health superimposed on progressively rising sea levels. The wide, low gradient continental shelf fronting most of this coast, particularly the Gulf of Mexico coastal states, significantly amplifies storm surge. There is growing confidence within the scientific community that threats from extratropical and tropical storm systems will increase under a changing climate. Increases in rates of storm intensification are already apparent. There is a broad consensus that already vulnerable communities in low elevation coastal zones (LECZ) will become much more vulnerable in future decades. As is clear from the *Sixth Assessment Report of the Intergovernmental Panel on Climate Change* [1], this increasing vulnerability results from a rising sea level [1–5], receding shores, warming sea surface temperatures, intensification of storms, extreme rainfall events, and socioeconomic drivers that amplify the physical impacts of the underlying climate-related hazards. Beyond environmental and socioeconomic drivers and consequences, public health questions, such as water-born health hazards, will necessitate increasing attention due to more frequent, widespread, and prolonged flooding events in LECZ. LECZ are extensive in both urban and rural areas of the southeastern U.S. states and are home to several immoveable coastal industries, many of which are tied to renewable and nonrenewable natural resources. These working coasts are important economic zones for the states and host critical infrastructure for military operations and national security, as well as large port facilities that serve as important nodes of the global economy. These zones are often near shifting sites such as coastal inlets or transgressive barrier islands. Despite the economic benefits presented by coastal industries, however, many LECZ residents are low income and underrepresented, resulting in increasing levels of social vulnerability to coastal hazards. Further, due to historical land use and development patterns, low income and minority residents in the LECZ are often physically the most at risk from these hazards.

The Fifth (previous) Assessment Report of the Intergovernmental Panel on Climate Change (IPCC) [6] considered *risk* to depend on *hazards, exposure, and vulnerability.* Coastal communities in the US and around the world are challenged by new and increasing hazards. As explained in a recent article by Viner et al. [7], risk is not static but dynamic and constantly evolving. Risk-informed decision making is crucial, but for the resulting information to be actionable, it must be effectively and promptly communicated to planners, decision makers and emergency managers in readily understood terms and formats. Observations and projections of human and physical factors that affect community vulnerability are essential to evaluate pre- and post-event conditions, to update baselines, to establish

objective model validations and to support both emergency operations and long-range regional planning. To objectively prioritize communities in the greatest need, it is essential that risk be quantitatively assessed taking account of all three of the risk factors described by the IPCC.

In the past, science-based decision-making tools and support systems have failed to adequately address the health and unique vulnerabilities of coastal residents in the southeastern U.S. LECZ, in part due to overly technocratic presentation and outreach programs in poor communities. Often, cultural backgrounds and social structures have been ignored, and this has prevented productive dialogues necessary for the development of meaningful solutions. In addition, most coastal locations are impacted by environmental factors that span multiple spatial and temporal scales [8]. Therefore, it is urgent to: (i) improve the understanding of the physical, ecological, health, economical, and social impacts of future flooding events and land loss trends; (ii) integrate the infrastructure and knowledge necessary to predict the future evolution of the human-nature coupled systems that define the southeastern LECZ; and (iii) create comprehensive decision support for local stakeholders within the southeastern LECZ that could serve as the blueprint for LECZ resilience elsewhere in the world. Stakeholders must be involved in two ways. First, stakeholders should participate from the start in co-developing understanding and knowledge. Second, each LECZ region should form a stakeholder advisory board that will ensure that stakeholder concerns and perspectives are considered at every level and in every decision. Lastly, a non-governmental, non-profit organization is needed to facilitate collaboration among performers from the public and private sectors. Effective, non-conflicted collaborative leadership of multi-disciplinary, multi-institutional teams is essential to innovation and solving complex societal problems.

Designing innovative methods to predict and prepare for future threats reaches well beyond traditional disciplinary boundaries. The right approach will involve multi-disciplinary and multi-institutional teams that include stakeholders, scientists, health professionals, emergency managers, politicians, and collaboration leaders, all with the depth of knowledge necessary to generate new emergent understandings of the interconnections of socio-economic, human health and natural systems. These understandings must include stakeholder perspectives and be effectively and readily communicated to non-scientists and the public in non-technical language. A major component of future efforts must include information dissemination via community education, cyber tool kits for managers, and the development of teaching materials for users, decision makers, and the public including coastal residents with little or no science background. Materials for public dissemination should include illustrated hard copy brochures in English, Spanish, and Creole (Patois).

An important vision must be to enhance future resilience of LECZ communities, not only in the USA but globally. Effective communication of information and understanding to residents and officials is essential to empowering citizens and businesses to effect change. Persuasion will depend heavily on how well the scientists and policy makers are able to grasp the socioeconomic drivers that cause residents or businesses to resist change. Strengthening communications and understanding among scientists, stakeholders, and policy makers is a fundamental goal advocated by this paper.

## 2. Research Approach

The purpose of this article is to offer a review and synthesis of recent literature on the matters described in the introduction and recommend possible approaches to identifying and addressing critical needs. The emphasis is on integrating natural and social sciences and facilitating a cyber-supported network of scientists, modelers, engineers, educators and stakeholders from academia, federal state and local agencies, non-governmental organizations, residents, and the private sector. This review resulted from a coastal resiliency initiative promoted and coordinated by the Coastal and Environmental Research Committee of the Southeastern Universities Research Association (SURA), in Washington, DC and

followed two SURA-led workshops hosted by the University of South Florida in Tampa and St. Petersburg, FL. Discussions at these workshops and during subsequent virtual meetings brought together experts from the multiple disciplines identified as critical to our goals. The team that emerged includes experts in the fields of social science, ocean science, numerical modeling, coastal ecosystem science, health science, environmental management, and policy and disaster planning. All members of this collaborative team offered their expert perspectives and experience in assembling this review. However, unlike most traditional multi-disciplinary reviews where each discipline is treated in its own sub-section, this team was tasked with assessing the future needs regarding transdisciplinary intersections with a high emphasis on engaging and communicating more effectively with non-scientific stakeholders and policy makers.

In addition to the informed opinions of the numerous co-authors, this collaboratively written narrative review considers recent literature from 1999 through 2021 but concentrates most heavily on papers published subsequent to the 5th Assessment Report of the Intergovernmental Panel on Climate Change (IPCC) which appeared in 2013. In other words, the emphasis is on works of the past eight years. This is not intended to be a systematic review but is more akin to a white paper aimed at identifying needs, research gaps, and the most urgent priorities for coping with the threats that climate change is likely to bring to LECZ communities in the Southeastern US as well as to similar environments elsewhere in the world. In what follows, we will begin with a review of some literature on social vulnerabilities and ways to improve effective communication with citizens and stakeholders, then consider the hazards, risks, and potential resilience of LECZ communities followed by a review of some existing and potential adaptive strategies. It was a consensus of our team that understanding "tipping points" where small changes in forcing can bring about dramatic shifts in natural or social systems is critical to predicting and adapting to future threats and we discuss this next followed by descriptions of four representative case studies of southeastern US LECZ. Finally, we review some crucial needs for building capacity to enhance LECZ resilience and highlight some key conclusions and recommendations.

## 3. Understanding and Engaging Vulnerable LECZ Communities

Over the past few years, there have been significant advances in understanding and modeling societal factors that can impact community resilience [9]. As the sophistication of geophysical models and flood hazard characterization [10] has increased, so too has our understanding of how frequently and severely episodic flood events and sea level rise (SLR) will impact citizens according to socioeconomic factors such as age, income, health, education, and ethnicity. The resiliency of a given community and its people to a natural hazard event depends on the geography of the location; the infrastructure of the built environment; economic vitality (e.g., sustainable livelihoods, income equality, employment rates); community health; social and human characteristics (e.g., social dependence, education, income, age, language skills); and population growth and mobility [11–13]. The concept of resilience is generally defined as: the ability to absorb change [14], the capacity to adapt existing resources and skills to new situations [15], the ability to be flexible and adaptable after environmental shocks and disruptive events [16], or the ability to resist, absorb, recover from, or adapt to an adverse occurrence [17]. One social resiliency approach uses an integrated demographic model [18] where community information is examined across demographic categories (gender, race, age, income, education) and by location and likelihood of a hazard event. Community modeling allows for the incorporation of stakeholder knowledge and adds context to geographic conditions. Areas having concentrated indicators of community vulnerability (or a lack of resiliency) are disproportionately prone to the social and economic impacts resulting from exposure to a natural hazard [19]. The sustainability of many coastal cities, particularly coastal megacities in poor countries, is already being measurably diminished by climate change [20]. To protect local economies and improve community resilience to coastal land loss, storm surge and compound flooding events we must improve access to, and the usefulness of, information available to

emergency managers and planners, as well as the public, so that all stakeholders are better equipped to supply, share, and act upon information in meaningful ways.

According to a recent Pew Research Center poll conducted in May of 2020, "A majority of Americans say they see the effects of climate change in their own communities and believe that the federal government falls short in its efforts to reduce the impacts of climate change." States themselves are not united regarding how to respond to a rapidly changing climate and local leaders often contemplate similar projects or studies without joining forces [21]. Localities find themselves operating alone or in regional consortia within state boundaries (e.g., Tampa Bay, where six counties are collaborating). State and local governments always have more budget demands than available funds [22]. The exigencies of immediate concerns push actions that address uncertain or even likely future events to the back burner [23]. Here, the grand challenge is not only to generate a more sophisticated understanding of the complexity of coastal change, but also to better educate and inform citizens about their role(s) in advocating for greater resiliency and identifying adaptations that people are willing to support. For policy makers, investments in coastal resiliency in response to projected future events (e.g., SLR, predicted increases in storms, harmful algal blooms) are much more difficult to sell to the voting public than investment in citizens' immediate needs and keeping taxes low. Citizens make more accurate assessments of their resiliency and better choices regarding hazard adaptation if the information they receive is communicated in ways that facilitate their ability to identify actions that preserve not only their lives, but also their livelihoods (employment), access to affordable housing and transportation, and social connections (family and community).

Better informed citizens can proactively improve their community's resilience in partnership with emergency management professionals, local officials, government agencies, and other key stakeholders. What is needed is a sophisticated assessment of the types of information, communication methods, and modalities of engagement that resonate with those vulnerable to coastal change. Communication and messaging preferences will vary across geographic locations and community profiles. The scientific community should build a library of communication strategies that can be adapted across localities using trusted information networks and familiar information channels or pathways, and messages that emphasize actionable information and communicates the social and economic impacts of impending and future hazards in ways that are relevant to the daily lives of the target audiences [24]. Effective communication outreach requires: (1) the utilization of trusted information networks (e.g., local or community groups that work directly with citizens and have population-specific expertise); (2) reliance on familiar information channels or pathways for target audiences (e.g., specific social media platforms, Spanish language outlets, or the voices of trusted neighborhood leaders), and (3) tailoring of communications to emphasize actionable information that avoids over generalization, but instead communicates the social and economic impact of flooding events in ways that are relevant to the daily lives of the target audience(s). Engagement with the community will enfranchise the voices of the those who have the most to gain or lose and will also build capacity of non-profit organizations. Central to LECZ resilience is the demonstrated engagement of local stakeholders regarding the consequences of the convergence of the natural and human environments due to climate change and SLR.

## 4. Assessing Coastal Hazards, Risks and Resilience

Climate change is increasingly being accompanied by the combining of multiple hazardous processes to create "compound events" [25,26]. However, traditional risk assessment analyses only consider one driver or hazard at a time and neglecting the importance of compound events typically leads to risk being underestimated. For the specific case of coasts, a major and increasing threat is related to compound flooding by storm surge and torrential rain [26], particularly as drainage systems in the LECZ lose capacity to rising sea level [27]. Analyses [28] also show that the occurrence of compound coastal flooding has been increasing over the past several decades. Educating politicians, public officials,

and stakeholders about the rising threats of compound flooding should be high on the list of priorities of any coastal resilience information network. Unanticipated flooding in recent years was caused by intense rains that accompanied Hurricanes Harvey, Irma, Maria, Florence, Dorian, and most recently Ida in August 2021. Compound flooding can also result from fluvial floods coinciding with storm surge particularly in river deltas [29]. The coupling of hydrology models to coastal ocean models is an area of active research and considered by some agencies as a grand challenge [30].

For the cases where flooding is the result of storm surge and/or waves alone, there are numerous reliable and well-tested predictive models of varying degrees of sophistication to choose from depending on local and regional bathymetry and shoreline complexity [31,32]. Storm surge models in use today include two-dimensional (e.g., vertically averaged properties across a spatial domain) and three-dimensional models using both structured and unstructured (irregular) grids. One of the simplest and lowest resolution models is NOAA's operational two-dimensional SLOSH model. SLOSH is the acronym for sea, lake and overland surges from hurricanes. The academic community uses more accurate unstructured grid models of coupled surge-wave effects (e.g., HYCOM, ADCIRC and FVCOM and DELFT3D). Although those models yield better results than the operational, long-standing two-dimensional SLOSH model used by NOAA for several decades, SLOSH continues to be NOAA's operational model of choice since it is well accepted, fast and does not require high performance computing (HPC, or advanced computing) resources. However, it is important for future applications to complex coasts, where street-level predictions are needed by emergency planners, for more accurate models to replace SLOSH as the model for routine operations. Recent advances in modeling include data assimilation from sources such as ocean observatories, nanosatellites, and social media. The most serious challenge lies in coupling the selected models with models of other crucial phenomena such as hydrology, erosion, ecology, and human attitudes and vulnerability.

The ability of a community to recover from a flood event depends more on local factors than on regional or global factors. For example, drainage infrastructure determines how long contaminated flooding waters may linger in neighborhoods and urban areas. Lingering floodwaters elevate risk due to physical hazards, increase property damage, and prolonged contact with disease agents, displaced animals, and toxic chemicals common in stormwater. The adverse consequences of flood events, especially coastal flooding, to human health have been evaluated by the World Health Organization [33] and the U.S. Global Change Research Program [34]. To properly address health-related issues, public health professionals, such as epidemiologists, must be involved in long range planning. The interconnections of water infrastructure (e.g., drainage), public health and sea level rise require careful consideration and perhaps a complex modeling approach [35]. Post-event recovery plans need to include detailed monitoring of water depth and water borne pathogens. It is possible that some of the monitoring could utilize trained "citizen scientists".

Many LECZs, where flood waters may advance and recede, have existing housing stock built prior to more-recent best practices or floodproofing regulations, ordinances, or building code provisions. The presence of residential structures (indeed entire neighborhoods and communities) on the floodplain may increase the rate of flood water rise, increase the flood elevation, and impede waters from receding. Claims history from the National Flood Insurance Program (NFIP) within Special Flood Hazard Areas (SFHAs), for example, demonstrate that at-risk areas may be subject to repetitive loss [36]

Whereas no areas within an LECZ are completely immune from risk stemming from natural hazards, some built environments are more at-risk relative to other areas [37]. It has been noted that low-lying areas near coastlines and rivers, over the course of several decades, may be subject to repeated loss due to a shorter return period between flooding events [38,39]. When repair and reconstruction take place on these damaged properties, largely in the same fashion and under the same standards as before the loss, this may perpetuate a cycle of continued damage and repair [40]. Focused on the longer-term reduction in risk, mitigation actions are intended to disrupt this pattern of repetitive

loss [41,42]. Indeed, structural approaches to manage floodwaters, such as the construction of levees, may actually foster development in these areas which may later flood when there is a structural or capacity failure in the water management system [43].

Despite well documented risks associated with coastal erosion, the consequences of SLR and compound flooding, and the advancements made in emergency warning systems and mitigation protocols, citizens often underestimate the extent of their own hazard risk [44–46]. Local leaders must operate in situations where individuals have historically made choices about where to live and work that are seemingly counterintuitive to the information available to them. The academic and practitioner communities have access to numerous resiliency assessment tools [47] such as the Social Vulnerability to Environmental Hazards Index [48] and geographic information system-based mapping applications such as the Federal Emergency Management Agency's (FEMA) HAZUS tool and the Virginia Flood Risk Information System (VFRIS) (https://consapps.dcr.virginia.gov/VFRIS/. Accessed on 27 October 2021). Flood Factor (www.floodfactor.com, accessed on 28 October 2021), a free online tool created by the nonprofit First Street Foundation, includes a searchable database of the current and future risk of flooding for any physical address in the U.S. First Street's calculations indicate that many cities have tens of thousands of properties facing risks not shown on federal maps and that minority residents represent a greater share of unmapped flood risk. Hidden risk lurks along the nation's coastlines. In Fort Lauderdale, Florida, FEMA estimates "about 41% of the city's 55,000 properties are located in the LECZ" but application of Floodfactor suggests that the figure is closer to two-thirds [49]. The U.S. Army Corps of Engineers (USACE), Engineer Research and Development Center (ERDC) is developing a tiered set of coastal resilience metrics that integrate engineering, environmental and community factors [50]. Selection of the most appropriate adaptation strategies will depend heavily on the details of the hazards themselves. The time scale of vulnerability is also important. Long-range (e.g., years to decades) vulnerabilities to gradually rising sea levels are typically very different than short-range vulnerability to an impending hurricane and storm surge [51].

Although predictive modeling of physical hazards such as storm surge are typically more advanced and quantitative than criteria for assessing the societal vulnerabilities to these threats, considerable progress in recent years has resulted in improved understanding and modeling of societal factors and changes that can impact community resilience [52–54]. It makes sense to assess and map the areas or neighborhoods of greatest vulnerability prior to predicting the distribution of hazards. The concept of baseline resilience indicators for communities has been more recently evolved as empirical metrics for gauging the resilience of communities to disasters [48,55]. Several researchers have developed social vulnerability indices (SoVI) that can be readily applied to most regionally specific communities and cities for purposes of planning for flooding related to climate change as well as for disaster management [56–59]. One of these indices [59] considers 15 different factors obtained from census data, most notably income and socioeconomic status, age and disability, minority status and language, type and quality of housing and access to transportation. To better support decision making and planning at the local level, risk perception should be included in models as a quantitative measure of community resilience. This approach allows for the modeling of strategic decision-making through assessments of the relative risk of alternative scenarios [60]. The existing PEOPLES framework [61] can be used to evaluate the resilience of a community and the "dimensions" that represent groups of interwoven societal, technical, economic, and organizational factors that serve as the foundation for quantifying community resilience given future hazard scenarios. The PEOPLES framework allows for assessing changes over time across the acronym's dimensions of: population demographics, environmental ecosystems, organized government services, physical infrastructure, lifestyle and community competence, economic development, and social-cultural capital.

## 5. Strategies for Adapting to Future Flooding, Land Loss and Health Threats

The coastal zones of the Southeastern U.S. are all highly susceptible to inundation, erosion, land loss, damage to buildings and infrastructure and harmful impacts on human health. The latest (6th) IPCC report [1,2] concludes that "It is very likely to virtually certain that regional mean relative sea level rise will continue throughout the 21st century, except in a few regions with substantial geologic land uplift rates. Due to relative sea level rise, extreme sea level events that occurred once per century in the recent past are projected to occur at least annually at more than half of all tide gauge locations by 2100 (high confidence). Relative sea level rise contributes to increases in the frequency and severity of coastal flooding in low-lying areas and to coastal erosion along most sandy coasts (high confidence)" [1] (p. 33). However, since socioeconomic, ecological, and built environment vulnerabilities vary by locality, each region requires its own unique assessment. In addition to variations in exposure to hazards, long-range vulnerabilities can be determined by a community's ability or willingness to adapt or relocate or by the availability of funds to build protective structures such as levees and flood gates. Short-term (days to months) vulnerability more often depends on health and mobility of residents, adequacy of evacuation routes, proximity of storm shelters and awareness of the public about the threats. The inputs from models of future hazard scenarios and the knowledge and opinions of local stakeholders, are required to assess the tradeoffs among multiple factors in terms of change thresholds or tipping points that could lead to new steady states.

Convergence of numerical modeling and stakeholder engagement can enable quantitative assessments of how coastal adaptation projects can alter the degree of resilience possessed by local communities or regions. Iterative interaction with the community is critical for identifying the most important factors and their correlations (such as the complexity and cost of change), as well as for modeling the range of individual responses to different disruptive scenarios. The framework and its results can be used as a decision support tool by stakeholders for selecting the optimal adaptation strategies that enhance a community's ability to thrive despite climate changes that increase the likelihood of disruptive events. The goal is the identification of tipping points where stakeholders and citizens become willing to support adaptive behaviors or policies (small and large changes) that promote resilience. Directly engaging community members in the scientific process and incorporating their local and traditional ecological knowledge into the numerical models elevates local knowledge through participatory modeling and directly addresses the complex challenges of climate-sensitive flooding and related impacts in areas with vulnerable populations [62,63].

Adopting practices that make LECZ homes less likely to be damaged by severe weather events makes sense, whether the approach is a structural adaptation to the structure, community-wide non-structural flood protection measures such development regulations and codes [42,43,64], or nature-based such as marshland restoration and living shorelines [65]. Homes that are more resilient will necessarily have less exposure and suffer less physical damage and property loss [66]. Although these more-resilient homes may not necessarily escape damage altogether, the extent of damage in these homes may be less due to these mitigation investments. In addition, resilience investments can reduce disaster costs [67]. Reducing potential damage to the structure by way of resilience practices has important follow-on consequences. First, households that may have been displaced may remain in the home, even if some repairs are required. Second, the financial burden of repairs may be lessened, reducing a significant burden especially for low-to-modest income households. Third, even when damage displaces a household, the repair and recovery time for the household may be shortened by resilience measures. Hastened reoccupation of the structure can reduce follow-on costs associated with prolonged displacement.

Hazard mitigation actions, such as those associated with land use planning and building ordinances, are generally focused on a reduction in the longer-term risk posed to structures and populations. The concept of severe storm hazard mitigation from the individual homeowner's perspective, includes the adoption of building practices intended

to reduce future risk stemming from flooding [68]. The development of mitigation strategies and practices, promulgated through regulations, ordinances, and building code provisions and government action, are intended to reduce the risk posed to structures within these flood-prone areas and, by extension, decrease damage, claims, population displacement, and pain and suffering. For many cities and towns, improved stormwater management practices and infrastructure are urgently needed. emphasis is placed on water-related infrastructure in coastal cities. Water supply lines are not too vulnerable, but sewerage is very vulnerable to tidal and storm inundation. To compound the problem, in urban areas with combined sewer overflow (CSO's) (https://www.epa.gov/npdes/combined-sewer-overflows-csos accessed on 28 October 2021) as in many older cities, runoff from storms further complicates recovery and can present serious post-storm health hazards.

Community workshops, outreach campaigns, and surveys allow residents and businesses a voice in assessment of the risks and rewards associated with measures that enhance resiliency including possible investments in infrastructure, changes to land use zoning policies, and the perceived benefits and costs of relocation. The cascading nature of decisions regarding resiliency necessitates frank discussions of the costs associated with short and long-term decisions. In concert with regional planners, local officials, and community organizations, regional teams can use the citizen survey software platform *MetroQuest* to connect with residents and local groups to assess risk tolerance and the kinds of adaptations stakeholders are willing to accept given SLR and flooding events, or the resiliency trade-offs that citizens are willing to support. Culture, values, scale, and context are key determinants of adaptation and adaptive capacity but are not universally recognized in local policies or by decision makers. The application *MetroQuest* can facilitate the assessment of what investments and/or trade-offs residents and business owners will support in terms of adaptive strategies (e.g., changes in regulations and practices in land development, infrastructure design and placement, and the source of funding for such endeavors. Environmental competency groups (ECGs) provide an avenue for democratizing the production of scientific knowledge [69]. The ECG method encourages collaboration between community members and scientists regarding hazard modeling and the design and implementation of coastal adaptation projects that are the most likely to be supported by key stakeholders including residents, business owners, and public officials. Policy makers will benefit from this information as it will allow them to focus resources (time and money) on those policies/strategies that stand the greatest chance of success. Unlike metropolitan areas such as Miami, many small cities and localities lack sufficient staff, expertise, and funding to be able to sustain community engagement. Robust non-profit organizations are critical to the efforts of local governments and agencies to develop and successfully implement policies that support resilient and vibrant neighborhoods.

Adapting to SLR requires both medium-term and long-term strategies. In addition to global SLR, there are also sea level fluctuations at multiple time and space scales that are more local or regionally specific [70]. Effective regional adaptation strategies typically involve extensive planning, large investments, and persuasion of vulnerable residents through outreach and education. Extensive modeling must precede implementation of strategies and, for the models to be trusted, they must be exhaustively tested against data and results must be communicated appropriately [71]. Different regions and agencies use differing modeling capabilities to forecast severe weather and ocean events. The strategies required to prepare, protect, or resettle threatened or displaced communities and help them adapt to changed or new environments is a major challenge that requires trust of officials and careful communication involving community leaders, including clergy, news media, community organizations and social media influencers. Some approaches are addressed in the multi-national *Ocean Obs 19* review paper on this subject [32]. Depending on how risk is defined, the number of people who may need to relocate varies between 88 million and 1.4 billion with most living in the LECZ or the 100-year flood zone of major rivers [72]. Hauer et al. [72] (p. 30) emphasize that: "migration from SLR is multifaceted, influenced by environmental hazards and political, demographic, economic and social factors em-

bedded within policy incentives to encourage or obstruct migration—not just SLR itself." Relocation, or even gradual retreat, are usually considered to be the strategies of last resort. Protection, (e.g., by seawalls or levees), or accommodation, (e.g., by elevating buildings) are less stressful on communities but may be less effective or more expensive depending on the circumstances. More shared data are necessary to understand the point at which individual choices. Such as relocation, become more attractive than remaining in place absent a disaster event. Important factors that need to be considered include: (i) what specific information resonates with citizens regarding their vulnerability; (ii) how do citizens weigh that information relative to perceived social and economic constraints (community and employment); and (iii) at what point does relocation become more attractive than enduring increasing frequent inundation and pollution. Some insight may be gained from examining communities in Bangladesh and low-lying Pacific islands (e.g., Marshall Islands), which are already yielding numerous SLR "refugees." In assessing the future likelihood of communities being relocated, social and economic factors commonly outweigh environmental factors. For example, in southern Louisiana, economic opportunities play a greater role than flood risk in influencing residents' decisions to move or stay [73].

The most formidable obstacles to relocation are economic, cultural, religious, or ideological mindsets. A prominent example of this is the refusal of the inhabitants of rapidly vanishing Tangier Island and Smith Island in the Chesapeake Bay to accept government funded relocation offers due to ideology-based denial of climate change and SLR [74]. The relocation of most Native American residents of Isle de Jean Charles, Louisiana to Houma, Louisiana [75] was somewhat successful in part since the community was kept intact, and the new location was relatively close to the old one. However, cultural traditions, morays and ideologies can impose serious resistance to climate change adaptation, particularly where indigenous people are involved [76]. Indigenous tribes may have divergent perceptions, risk tolerance and preferences for adaptation versus relocation [77].

## 6. Identifying and Modeling Tipping Points in LECZ Systems

Physical, environmental, economic, and social changes in coastal areas interact, accumulate, and occasionally, reach a *tipping point*—a point at which one more small change results in a large destabilization and transformation of the environment, such that it enters a new state [78]. Critical LECZ tipping points can be physical, ecological, or socioeconomic. In one coastal ecological example of this, the sea level rate of rise *tipping point* for delta accretion versus submergence was found to be about 5 mm/yr. [79]. Delta growth cannot keep pace with more rapid rates of SLR and open water will replace dry land. The concept of tipping points is increasingly being applied to assessments of the vulnerability and community resilience in LECZ. The concept of tipping points can be applied where communities along the coast are subject to a variety of mounting pressures resulting from ongoing coastal land loss and inundation. These pressures will determine for how long these communities can remain viable. For example, a frequent tipping point identified by residents had to do with the frequency and severity of flooding. The threshold is frequently tied to how often water gets in homes and the capacity of owners to deal with repairing damage financially and physically. Opportunities for employment were also cited as potential reasons individuals might consider relocating. Residents of a region in Louisiana point to economic factors and the ups and downs of coastal industries (oil and gas, in particular), and geographic location (i.e., proximity to oceanfront) [80] as important determinants of how individuals and families make decisions about whether to relocate. If the threats outweigh the prospect of jobs, individuals may be persuaded to relocate but if jobs come back to the coastal region this would be motivation to move back. In response to these pressures, local, state, and federal agencies have developed a variety of restoration plans and policies that decisionmakers can use to potentially extend or shorten the lifespan of these communities. Each region or community will have its own unique set of factors that determine tipping points for investing in infrastructure, relocating, or simply staying put and these factors should be well understood before adequate adaptation planning begins.

The concept of tipping points has been incorporated into formal frameworks for adapting to uncertain, evolving climate change impacts over long time scales. For instance, the dynamic adaptive policy pathways (DAPP) framework is a sequence of problem framing and analytical steps used to develop a roadmap of potential sequences of decisions that can be taken through time as risks evolve [81]. The approach relies on identification of tipping points that lead to physical or socio-economic conditions where existing risk management infrastructure and policies would be overwhelmed. This approach has been adopted in several high-profile applications aimed at adapting to sea level rise, including delta management in the Netherlands [82], the Thames estuary 2100 study [83], and flood risk management in New Zealand [84]. However, the identification of tipping points often relies on quantitative simulation models that may not be available in all contexts, particularly when financial resources and technical expertise are limited. Additional work is needed to determine how best these frameworks can be applied in more resource-constrained, local contexts [85]. In these situations, creative solutions are necessary to address some of the more technically challenging components of the DAPP process (such as exploratory modeling) and to find adaptive options that are financially and institutionally feasible.

## 7. Select Examples of Vulnerable LECZ Communities in the Southeastern U.S.

Evolving a national response to SLR and associated coastal inundation in the U.S. has proven difficult due to several factors including the high variability in regional and local factors, such as subsidence and ocean current behaviors, that interact to affect relative sea level. There is no "one size fits all" suite of solutions for the diversity of coastal regimes that exist in the Southeastern U.S. and the roles and behaviors of the numerous factors vary regionally and among different communities. To illustrate this, we have selected two prominent "case study" examples from the Gulf of Mexico coast and two from the Atlantic coast. The four cases were selected since they represent the range of diversity of southeastern US LECZ and since they have been extensively studied by various members of the multi-institutional team of this paper's coauthors. The Gulf coast regions are (1) coastal Louisiana's "bayou" region and (2) the west Florida coastal region extending from Tampa Bay north to Cedar Key. The Atlantic coast examples are (3) the southeast Florida counties of Miami Dade and Broward, and (4) the Mid Atlantic coasts of Virginia and Maryland centered on Chesapeake Bay. Each of these regions has its own unique set of physical, socioeconomic, ecological, and infrastructural circumstances.

Some of the major resilience-determining characteristics of the U.S. Gulf of Mexico coast are described in recent reviews [86–88]. The states of Florida, Alabama, Mississippi, Louisiana, and Texas all have extensive low elevation coastal zones (LECZ) bordering the Gulf of Mexico. The current population is over 15 million people (most of whom live in vulnerable LECZ communities). The northern Gulf Coast supports over 4000 oil and gas platforms. Tourism is the major industry along Florida's Gulf Coast. There are also numerous major ports in the region including Tampa Bay, Mobile, New Orleans, Port Fourchon, Houston, and Galveston. The entire Gulf coast lies in the path of major hurricanes and is also subject to flooding by torrential rains and rivers. The continental shelf fronting the Gulf of Mexico coast is extremely wide and gently sloping causing significant amplification of storm surges (Figure 1). Wetlands and the LECZ extend many miles inland from the present shore. Sea level rise scenarios [89] suggest that by 2050 the shores of many Gulf Coast shorelines could transgress inland several miles. Coastal Louisiana is the most threatened and this was dramatically highlighted by the August 2021 landfall of Hurricane Ida.

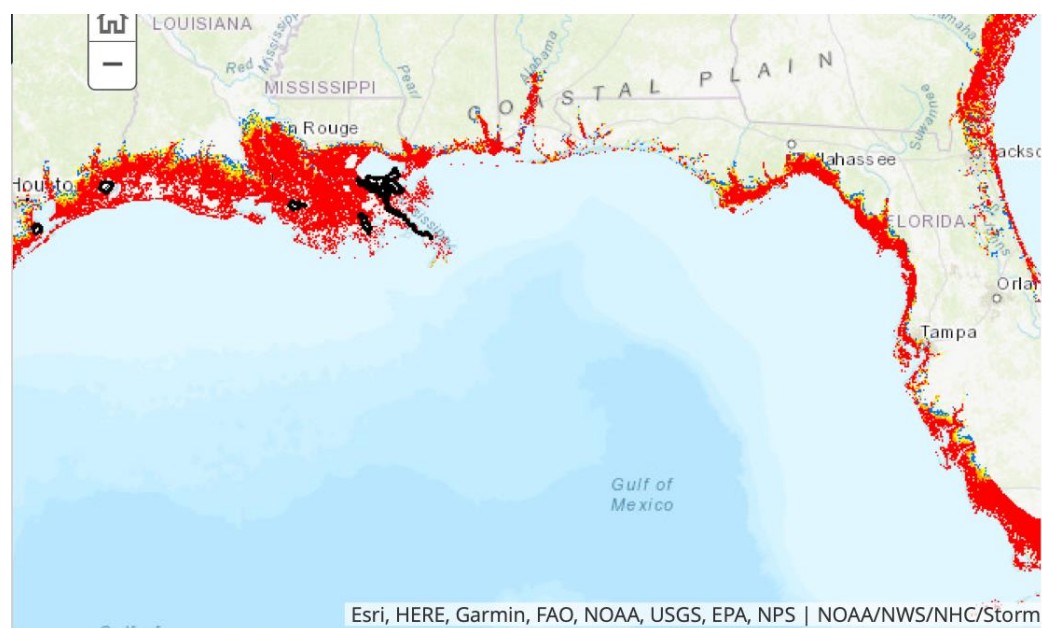

**Figure 1.** National Hurricane Center predictions of the "maximum of maximum envelopes of water" (MOMS) caused by storm surges associated with landfalling category five hurricanes on the central and eastern regions of the Gulf Coast. The red areas could expect inundation to exceed 9 feet (~3 m) above existing ground levels. Levees currently protect areas shown in black. This public domain image from NOAA was also presented in [31,86].

### 7.1. Example 1 Coastal Louisiana

The coastal Louisiana regional example embraces Plaquemines, Jefferson, Lafourche, and Terrebonne Parishes, along with Barataria and Terrebonne Bays. This includes locations such as Port Fourchon and the city of Houma as well as other major Louisiana ports such as the Port of Greater Baton Rouge, Port of South Louisiana, and the Plaquemines Port. Coastal Louisiana is home to the largest wetland in the USA and supports the nation's largest commercial fishery, supplies 90% of nation's outer continental shelf oil and gas, and facilitates about 20% of the nation's annual waterborne commerce. The region is also facing several environmental and climate-driven threats and the rate of land loss currently exceeds 41 km² (16 mi²) per year [89]. The Louisiana Coastal Protection and Restoration Authority (CPRA) [90] projects that by 2050, without restoration and protection, most of the wetlands will have been replaced by open water. Further, the Mississippi Deltaic Plain is subsiding at rates up to 18 mm/yr. [91]. When this subsidence is added to the projected rates of global sea level rise of between 8 mm/yr. and 16 mm/yr., the total relative rate of sea level rise in coastal Louisiana will conceivably reach between 26 mm/yr. and 34 mm/yr. or roughly up to 1 foot per decade. Many communities in coastal Louisiana face imminent pressures to relocate [75,92]. In summer 2020, Hurricane Laura, Hurricane Sally, Tropical Storm Beta and Hurricane Delta caused extensive flooding and serious damages in coastal Louisiana. Category 4 Hurricane Ida, in August 2021, was far more devastating to numerous coastal Louisiana communities including those in Jefferson, Plaquemines, and Terrebonne Parishes.

### 7.2. Example 2 Florida Gulf Coast

The other Gulf coast example is the west central Florida coast. The total population of the Tampa Bay Area, which includes the cities of Tampa, St. Petersburg, and Clearwater is 3.14 million people of which 14.6% are impoverished. Flood prone areas have both poor and wealthy neighborhoods. The Port of Tampa is Florida's largest port in terms of cargo tonnage and is the home port of six major cruise ship lines. The Port of Tampa, as with many other ports, is being impacted by SLR and increased storms. The World

Bank concludes that Tampa is among the 10 cities most at risk from climate-change related flooding. In terms of potential risk to human life and well-being, Tampa Bay is among the most at-risk coastal cities in the U.S. One of the reasons for Tampa's greater risk and exposure is that, in addition to storm surges being larger over wide, shallow continental shelves, as contrasted with narrow, steep continental shelves, the length and shape of the bay results in increasingly higher surge heights farther up the bay as the surge propagates up the bay [88,93,94]. NOAA predictions indicate that a landfalling category four hurricane would cause storm surges more than 20 feet high in the upper reaches of the bay, where the Tampa city center is situated (Figure 2). Considering how real hurricane winds might set up sea level throughout the bay [94], certain areas could see inundation exceeding 14 ft above ground level in the case of a category three storm such as Hurricane Ivan. These predictions do not include the potential added effects of compound flooding that would prevail if the storm surge were accompanied by prolonged torrential rains. Waves add destructive forces to the flooding effects of storm surge and severe rains [95] to destroy structures and other infrastructure.

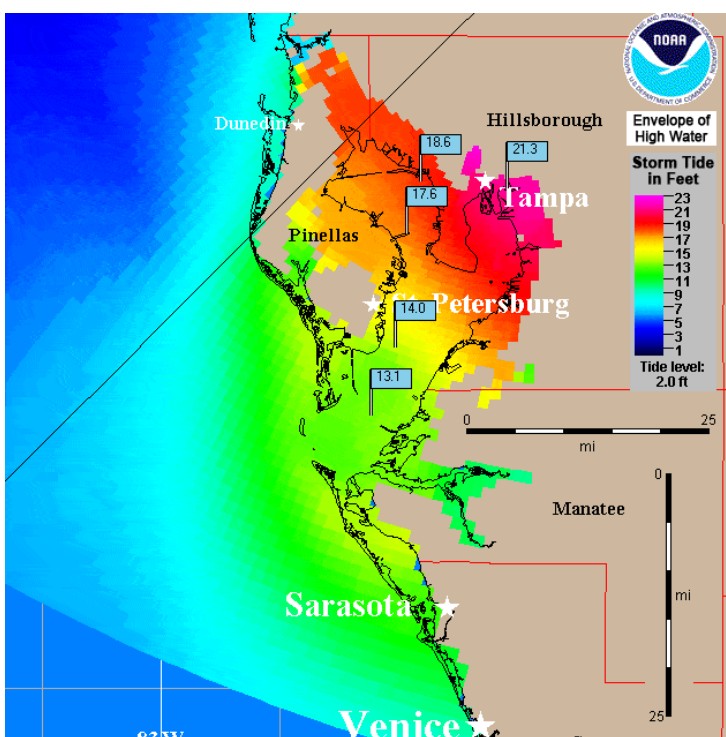

**Figure 2.** Predicted storm tide (storm surge superimposed on high tide) height for a category 4 hurricane making land fall on Tampa Bay at high tide. Due to the funneling effects of the bay, the areas in the upper (northeastern) part of the bay where Tampa's city center is located could expect over 20 feet of flooding. Model results from NOAA's National Hurricane Center utilizing the sea, lake, and overland surges from hurricanes (SLOSH) model. Source: Public domain NOAA image also presented in [87].

The Tampa Bay region includes considerable existing stakeholder activities. Along with the various agencies and NGOs (e.g., the National Oceanic and Atmospheric Administration (NOAA) Office for Coastal Management, the Alliance for Coastal Technologies, Tampa Bay Conservancy, local Audubon chapters and centers, the Tampa Bay Regional Planning Council, and the Tampa Bay Regional Resiliency Coalition http://www.tbrpc.org/resiliency/ (accessed on 27 October 2021) comprised of public officials and agency representatives from Citrus, Hernando, Hillsborough, Manatee, Pasco and Pinellas counties and 21 area municipalities), the USF Coastal Ocean Monitoring and Prediction System (COMPS) runs daily, automated nowcast/forecast models for Tampa Bay and the larger

west Florida continental shelf. In addition, the USF Center for Maritime and Port Studies (CMPS) collaborates with the USF Colleges of Public Health, Engineering, Business, and Global Sustainability. Additional entities include the National Maritime Law Enforcement Academy, Port Tampa Bay, Maritime Tactical Systems, Inc., Pole Star Space Applications USA, Inc., and Port Tampa Bay.

North of Tampa, Florida's rural and sparsely populated "Nature Coast" covers 4000 km$^2$ (1500 mi$^2$) and includes 8 counties on Florida's Gulf Coast [96]. Much of this area is water, small islands, or intertidal wetlands (Figure 3). Environmentally and socio-economically the contrast between the urban regions of Florida and the Nature Coast is truly extreme. Unlike the shores of much of Florida, the Nature Coast is not fringed by sandy beaches, but by brackish marshlands that grade almost imperceptibly into shallow open water bays, vital seagrass ecosystems, and, ultimately, the Gulf of Mexico There are no expensive resorts or high-rise buildings on the Nature Coast. Resources for supporting coastal resilience there are lean.

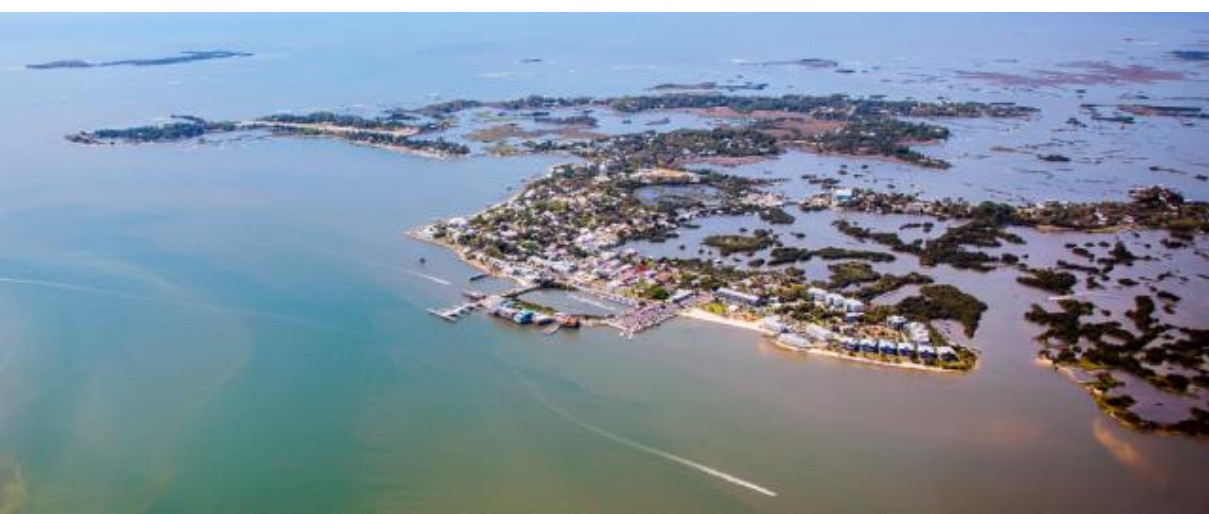

**Figure 3.** Cedar Key on Florida's Nature Coast, Levy County. Presented previously in [87].

*7.3. Example 3 Southeast Florida*

The southeast Florida Atlantic coast region contrasts sharply with the Gulf of Mexico coast region in many respects [9,96]. It is characterized by carbonate beaches, mangrove wetlands, low elevation, highly populated urban areas, a narrow and steep continental shelf, and susceptibility to tropical cyclone impacts even though storm surge amplification is much less pronounced than on the Gulf Coast. Natural coastal features such as coral reefs, beaches, dunes, and mangrove wetlands provide a natural adaptation to extreme weather events and help to limit erosion and flooding while also providing wildlife habitat and valuable recreational opportunities. However, the now coupled human-natural systems combines geological, environmental, and socioeconomic factors that lead to very high risk from short-term weather events and longer-term climate oscillations and trends. A factor that influences all coastal environments along the east coast of Florida is the variability of Gulf Stream flow, which drives short term coastal sea level shifts of 70 cm or more. Gulf Stream influence on coastal sea levels extends well to the north of Florida and is a factor in the sea level rise "hot spot" along the mid-Atlantic coast and northward [97]. Recent analyses [98] indicate that the Atlantic Meridional Overturning Circulation (AMOC), of which the Gulf Stream is a part, may be close to a tipping point for transitioning to its weak circulation mode. A major weakening or shut down of the Gulf Stream could result in significant sea level rise along much of Atlantic coast particularly south of Cape Hatteras and along Florida's Atlantic coast.

Miami-Dade and Broward Counties respectively cover 5200 km$^2$ (2000 mi$^2$) and 3186 km$^2$ (1230 mi$^2$). The major cities are Miami, Miami Beach and Ft. Lauderdale along

with numerous smaller cities. The combined population of Miami-Dade and Broward Counties was 4.5 million in 2015 and most of the people are Hispanic [99]. The urban infrastructure is built on low lying Pleistocene carbonate platforms having very little topographic relief. Human activities have highly altered southeast Florida hydrology dominated by porous limestone, infringed on, and destroyed large areas of wetlands and produced a mix of vulnerable densely populated coastal residential and commercial development. Southeast Florida includes two major commercial seaports, Port Everglades, and the Port of Miami that handle increasing amounts of commercial cargo. Both ports require continued improvement of infrastructure, adaptation to climate change, and consideration of security issues.

Key stakeholders in southeast Florida provide extensive support for coastal resilience. They include the Florida Broward County Environmental Protection and Growth Management Department, the Miami-Dade Sea Level Rise Task Force, the Palm Beach County Office of Resilience (OOR), and the Monroe County (Florida Keys) sustainability Office. Other major stakeholders in southeast Florida include the South Florida Water Management District (SFWMD), the Authorities for the Port of Miami and Port Everglades and the U.S. Navy's South Florida Ocean Measurement Facility (SFOMF) representing a major military presence in southeast Florida. The Florida Resilient Coastlines Program under the auspices of the Florida Department of Environmental Protection (FDEP) provides guidance and small grants to individual Florida communities on a competitive basis for improving coastal resiliency and adaption to sea level rise. One of the major future needs for Southeast Florida is to interlink the coastal resiliency efforts by providing user friendly access to data and analyses that cross-cut local and county boundaries for an integrated approach to manage coastal hazards and plan for long-term adaptive management.

### 7.4. Example 4 Virginia and Maryland Coastal Region

The Virginia-Maryland coastal region is in the approximate center of the Middle Atlantic Bight (Figure 4). It embraces the Chesapeake Bay and includes multiple urban centers (Baltimore, Washington, D.C., Richmond, Norfolk, Virginia Beach and Hampton Roads) as well as large rural and agricultural areas. Unique among the four regional examples described here is this region's vulnerability to extratropical (Nor'easter) and tropical storms as well as hybrid storms such as 2012's Hurricane Sandy [100]. This region provides examples of both urban and rural areas, as well as areas with both political and economic importance. The cultural and political importance of Washington DC is matched to a great extent by the importance of Hampton Roads to both US national security and the regions' tourism industry. Hampton Roads, which has a population of 1.7 million, is routinely exposed to flooding by hurricanes and Nor'easters as well as by fair-weather "nuisance flooding" and is experiencing one of the largest rates of local SLR in the U.S. [100]. Studies have shown that Hampton Roads and other East Coast cargo container terminals have differential risk to sea level, storm surges, and tidal flooding, with coming decades showing heightened susceptibility of the terminals and supporting transportation and utility infrastructure [101].

Storm surges are often more severe than in southeast Florida due to the wider continental shelf and the funneling effects of the Chesapeake Bay. As in southeast Florida, aperiodic inundation events result from reduction in Gulf Stream transport among other non-storm factors [97]. Land losses on Smith Island and Tangier Island in the Chesapeake are accelerating and may soon force the relocation of long-term residents. Many of the barrier islands on the Eastern Shore of Maryland and Virginia are experiencing landward transgression [102].

The Virginia/Maryland region already hosts stakeholder workshops and the twice-annual 'Rotating Resilience Roundtables' [103–105] that bring together a network of stakeholders, including Virginia's planning districts, local officials, and nonprofits such as Wetlands Watch. The region benefits from Virginia Sea Grant's extensive stakeholder network. The Roundtables help to facilitate interactions between academic and non-academic

stakeholders to improve the alignment of science with local problems in different coastal communities" and "identify pressing issues and knowledge gaps." The priorities identified by Virginia stakeholders during its Rotating Resilience Roundtables in 2018 and 2019 [103,104] were: *"It is important to ensure that resilience strategies reflect [the varied] contextual characteristics of communities (cultural, historical, and socioeconomic)"* [103] (p. 17), and *"There is a need to identify ways in which forests can contribute to flood risk reduction and how forest conservation can be advanced to support coastal ecological resilience"* [104] (p. 7).

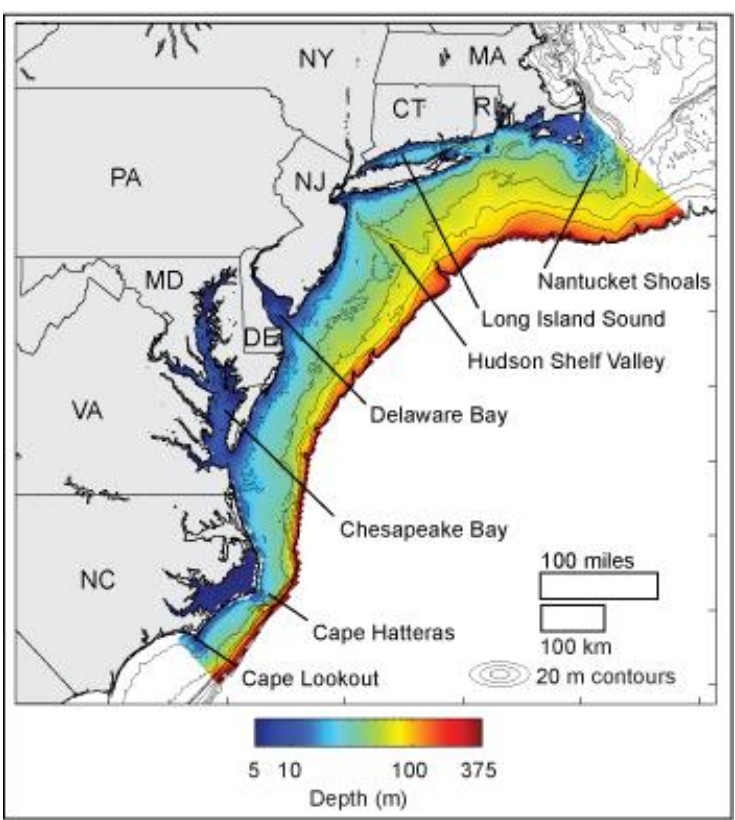

**Figure 4.** The Virginia-Maryland coastal region is roughly in the center of the Middle Atlantic Bight and includes the Chesapeake Bay. The width of the continental shelf here is 125 km. Source: U.S. Geological Survey Public Domain Image.

## 8. Transdisciplinary Capacity Building to Support Vulnerable LECZ Communities

Fortunately, all four of the LECZ regions described in the previous section possess extensive resources and state or county agencies to provide reasonable levels of resilience support, at least in the near term. Many other more sparsely populated regions and low-income communities in the southeast U.S. are less fortunate. Despite the diversity of physical, ecological, and socioeconomic drivers of vulnerability among regions, there are some clear needs for building the capacity to endure and adapt to future climate-induced changes in coastal threats. Consistent with conclusions recently articulated for future coastal science and engineering endeavors [105,106], we need to *identify, understand, predict, and communicate the interconnectivity of complex coastal systems including man-made aspects [e.g., ports, transportation infrastructure], the natural coastal environment, and socioeconomic systems (community vulnerabilities) in response to flood hazard events and long-term sea-level rise.* The capacity to do this must extend well beyond dealing with short-lived storm events to include decadal scale long-term changes in threats, risks, and consequences. Doing this will require extensive and iterative collaborations among diverse teams of physical, ecological, social, health and data scientists; policy makers; resource and emergency managers; and the public at large. In addition to extensive numerical modeling and observational monitoring, an accessible and ever-expanding data base and the support-



ing cyber infrastructure and a fluid bidirectional dialogue among scientists, policy makers and citizen stakeholders will be critical at a hierarchy of local, regional, and super-regional scales. Success of future coastal resilience programs will be critically dependent on the effectiveness of communications between scientists and non-scientists [24,107].

Resiliency programs must be based on options that are attractive to citizens, and models must explicitly account for the attitudes and concerns of business owners and residents and prioritize education and outreach to affected communities and businesses. Workshops must be employed to better understand what drives decision-making and how to address resistance to change and encourage behaviors that enhance resiliency. Scenarios and data-driven demonstrations enable stakeholders to improve their decision-making processes with new technologies. Decision aid toolkits should include dashboards that can be viewed from mobile devices. Future programs should aim to: (1) advance our understanding of stakeholder risk perceptions and community tolerance for hazard adaptations; (2) advance our understanding of how coastal change affects the built environment (and vice versa); and (3) improve the content and delivery of risk communications that advance resiliency based upon best practices. Future communications of science to stakeholders must emphasize actionable information about the social and economic impact of flooding events in ways that are relevant to the daily lives of the target audience(s).

For regional programs to reach the capacity to adequately accomplish the general goal set out above, there are several specific knowledge gaps that must be addressed within the contexts of regionally specific sets of environmental and socioeconomic factors. Some of these knowledge gaps have been described by government agencies and nongovernmental organizations (NGOs). For example, the National Science Foundation Coastlines and People program and the NOAA Coastal Resilience Grants program are focused on applying research advances to improve resilience. NGO programs such as the Pew Charitable Trusts' flood-prepared communities initiative, The Nature Conservancy's Resilient and Connected Landscapes project, and Climate Central's program on Sea Level Rise have addressed research gaps by providing communities with local data and graphics. SURA and collaborators involved in coastal resilience from academia have put these research thrusts into context for Southeastern US LECZ and agreed, following extensive discussion, that the following gaps are the most critical:

1.  The impacts of coastal flooding exacerbated by population growth and sea level rise, among other stressors, are expected to cause concurrent change in the coupled natural-human system if unchecked. We should seek to ***identify, understand, and predict tipping points in the natural and social systems***, with particular emphasis on understanding complexities associated with loss of function of the built system, land-based economies, social justice, and human health. This issue is cross-cutting and of critical importance to identification of policy, planning, citizen science and engineering interventions that will help prepare for, prevent or delay catastrophic tipping points.

2.  Although it is widely accepted that coastal flooding hazard is accelerating, it is not yet known how people and communities will respond over the long term, particularly in the face of unknown changes to the hazards arising from compound events (e.g., high surge coincident with high precipitation). We should seek to ***understand and predict change in coastal flood risk perceptions and attitudes over time, and how differing place-based characteristics, land use/development decisions, policy, and infrastructure investments influence these perceptions and attitudes—and vice versa.*** For example: What strategies promote equitable adaptation? "In what manner...are decisions made in 'rebuild or leave' scenarios? What are the sensitivities to variations in factors such as resource allocation, recovery rate of damaged infrastructure, or the availability of critical facilities? [105]. Communication must be supported through familiar and trusted messengers to provide clear, actionable information, and tailored messaging and information pathways for target audiences [24,108]. Programs should develop tools to enable communities, planners, and decisionmakers to evalu-

ate regional (multiple communities, state-scale or larger) strategies in the context of promoting equitable outcomes across communities.

3. With sea-level rise and population growth, aging infrastructure is increasingly vulnerable to coastal flooding, posing risks to both public health and ecosystems. We should seek to *understand and predict the impacts of coastal flooding (chronic and episodic) on the release of contaminants into surface-water and groundwater, and its cascading impacts on public health and land-based local economies*, with particular emphasis on point-source and emerging contaminants such as the loss of integrity of underground storage tanks and septic tanks. For example: What natural-built-policy scenarios are least (or most) likely to lead to disparate impacts on various coastal communities, both rural and urban? What novel methods (e.g., remote sensing) can be leveraged to identify and characterize potential point sources? What innovative solutions (technological, organizational, and political) might help mitigate or even prevent contamination? Outcomes should include predictive tools that enable equitable (both in terms of public and economic health) decision-making.

4. Nature-based adaptations to climate change have the promise to protect against coastal hazards while delivering ecological, recreational, and other services. *For these benefits to fully manifest, we must first understand how design and decision optima are influenced by consideration of diverse, uncertain ecosystem services. Concurrently, we must ascertain how scientific advice is best developed with and delivered to vulnerable communities to support their diverse interests.* For example, vegetative communities differ in terms of support for both ecological function and ability to control risks of flood by dissipating kinetic energy of storm surges [108–110]. Nature-based adaptations therefore often pose tradeoffs across disciplines, for which there is a lack of integrated modeling capacity [111]. There are also important uncertainties regarding which scientific products or services best serve vulnerable communities in the setting of environmental decision-making heavily mediated by political processes. Diversity in the natural, built, socioeconomic, and political settings provide cross-cutting opportunities to investigate how conservation policy and green infrastructure implementation would play out over time. Models are needed that couple ecological and physical sciences within a decision-making framework and enable scientists and communities to communicate their interests to policymakers.

5. Coastal resilience imperatives have sparked numerous policies and land use/development, and infrastructure ideas emanating from all levels, from the individual resident to the government. *We should seek to understand and assess the validity of these many overlapping and sometimes contradictory ideas through elicitation, equitable evaluation, and numerical modeling. Further, we must identify barriers and conflicts that impede implementation of scientifically valid resiliency measures.* By engaging scientists as well as stakeholders representing intersecting economic, political, and community-based constituencies, we may uncover differences in how resiliency measures are identified (such as wastewater management, mitigating coastal erosion, storm surge protection, etc.), who promotes them, and the differential effects likely to be experienced by various stakeholders. For example: How can possible and desirable resiliency scenarios be developed in collaboration across multiple constituencies and interests? In cases where model results contradict stakeholder or policy maker opinions, we must strive to explain the conflicts in non-technical and easily understood terms. And scientists must listen carefully to stakeholder perspectives; models can be wrong.

It will be necessary to develop a network to assemble and distribute information, including model results, to assess risk and its causes and improve the ability of coastal communities to adapt to change. Interdisciplinary collaborations can empower and enhance the capacities of localities to effect change. Regionally specific research to address these critical knowledge gaps will require transdisciplinary teams, appropriate suites of numerical models, remotely sensed and directly monitored observations as well as cyber

infrastructure for storing and sharing data and model results and supporting model runs. Virtual platforms should be created for compiling and sharing critical information related to each of the six categories of understanding listed above (i.e., (1) societal; (2) health; (3) economic and infrastructure; (4) land loss; (5) ecology; and (6) inundation). Remote sensing data from geographical information systems (GIS), data archives, citizen scientists, and artificial intelligence will ultimately be needed for the purposes of adaptive management. Web-based platforms such as NSF's *DesignSafe* Cyber system can facilitate data and model sharing. The southeast should exploit the Integrated Ocean Observing System (IOOS) Regional Associations, the Chesapeake Bay Interpretive Buoy System, and the National Mesonet.

For the programs we advocate to succeed, it will be imperative for scientists and stakeholders to readily share data, predictions, social vulnerabilities, and changing needs and priorities via an easily accessed cyber network [112]. While data-sharing networks have existed for several decades, they have been largely intended for use by the *cognoscenti.* The network we envision must be underpinned by as much simplicity as is feasible but still be capable of supporting sophisticated modeling. The stakeholders are likely to only need access to clear, graphically displayed model results while the scientists will want to share open-source model code as well as observational data and bathymetric grids for model runs. In some cases, the questions sought from cyber infrastructure may simply require a map visualization of a "what if" inundation scenario. In other cases, a data-intensive application may be required to provide rapid dissemination of information to emergency managers. Data from geographical information systems (GIS), data archives, citizen scientists, and artificial intelligence will ultimately be needed for purposes of adaptive management [113]. Another important, but conventional, role for information technology teams is likely to involve facilitation of virtual workshops. Web-based platforms such as the *DesignSafe* Cyber Infrastructure component of Natural Hazards Engineering Research Infrastructure (NHERI) facilitate data and model sharing.

It must be emphasized, however, that having a network that enables the sharing and communication of information that is readily understood by the public does not negate the parallel, and essential, need for high performance computing (HPC) resources that can serve the needs of data-intensive analyses including artificial intelligence and advanced complex systems modeling. Such advanced capabilities will underpin future quantitative predictions of the many interconnections among the myriad physical, ecological, and socioeconomic factors that will govern future resilience of LECZ regions [113]. Application of complex, transdisciplinary community models that account for the often non-linear and multi-directional linkages among the social sciences, ocean sciences, atmospheric sciences, ecosystem sciences and coastal engineering will ultimately provide the long-range adaptive management guidance that will be needed [30]. It is understood, of course, that most LECZ regions or towns will not possess their own HPC resources. The concept is that these entities should be part of larger networks that are connected to centralized or super-regional hubs where HPC capabilities reside (Figure 5). These major hubs may be based at large universities such as the University of Florida which is home to the HiPerGator supercomputer (https://www.rc.ufl.edu/hipergator-ai-achieves-highest-top500-ranking-for-uf/) (accessed 21 October 2021)or at national facilities, such as the Department of Energy's (DOE) Thomas Jefferson National Accelerator Facility in Newport News Virginia or possibly at NOAA. The Thomas Jefferson facility is a component of DOE's Exascale Supercomputing Network (https://www.energy.gov/science-innovation/science-technology/computing) (accessed 21 October 2021). The ongoing progress being made with the growth of *Internet 2* may soon enable broad-band transfers of huge data sets and model outputs among large hubs and among the multiple nodes of regional networks. Clearly, the efficacy of such capabilities will rely heavily on the dedication of advanced teams of data scientists and information technology professionals. Obviously, with any computer network cybersecurity must be a serious factor in determining trust by stakeholders and these large HPC facilities must have advanced cybersecurity protocols. Networks must also implement rigorous

standards for the data that they accept to prevent misinformation from being loaded onto their platforms. This will require the major data hubs to have coastal and social scientists on their full-time staffs to ensure critical review of the information that is disseminated.

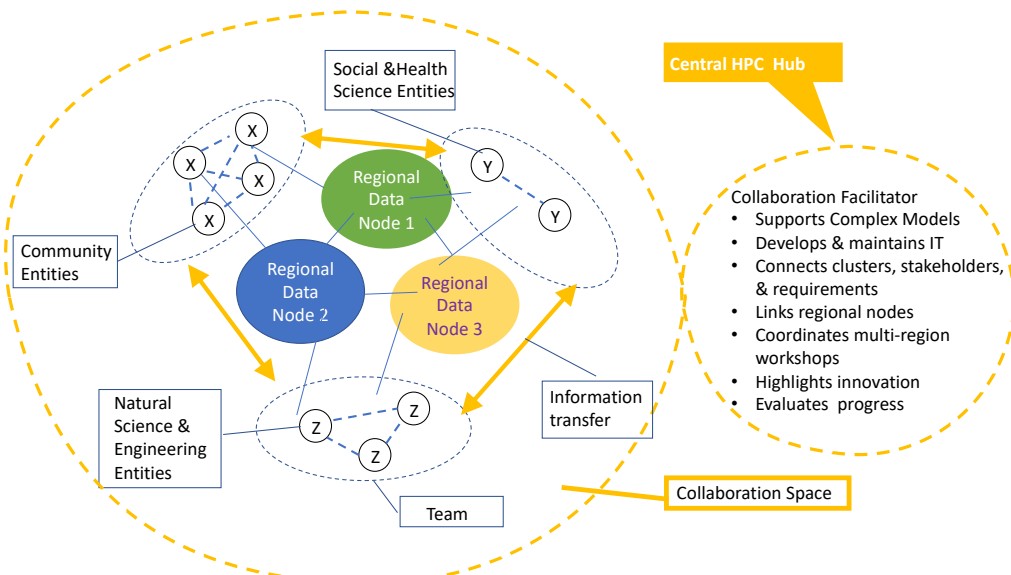

**Figure 5.** Concept of network integration involving regional nodes, a central HPC hub, talent clusters, and multiple disciplines with both regional and central (or super-regional) collaboration facilitation and management. Source: original figure.

## 9. Conclusions and Recommendations

Numerical models are essential to predicting future threats at multiple time scales from short to long term and support for their continued development and refinement should be ongoing. High-resolution models of specific processes such as storm surge will continue to underpin emergency planning. However, for many LECZs, the existing storm surge models will probably suffice for the next few years; some are already able to forecast inundation at street level resolutions [114]. ADCIRC or similar high-resolution models should hopefully replace NOAA's aging SLOSH model for operational applications soon. What are more urgently needed now are new models that connect ocean and hydrologic behaviors to predict compound flooding [26–30] at local, high-resolution scales and application of those models to specific vulnerable communities.

Beyond predictions of the physical processes alone, we need to begin linking these physical process models with social and health science models to better understand the potential impacts of anticipated climate changes on regions, towns, and neighborhoods. For adaptive planning to be effective, transdisciplinary collaborative teams should strive to map the intersection of the multiple physical, ecological, socioeconomic, and human health factors that are likely to bring about critical and unexpected tipping points in the foreseeable future. To reach the level of understanding and science-based planning that is needed for long-term LECZ resilience, data intensive and artificial intelligence analyses and complex systems modeling must ultimately become part of future coastal research programs at a hierarchy of local to global scales. As pointed out in the beginning of this paper, extensive collaboration will be critical to future successes. Future collaboration will rely on networks such as those illustrated in Figure 5 to connect observers, modelers, managers and first responders so that solutions can flow readily across disciplinary and stakeholder boundaries [32].

There are numerous threatened LECZ regions and communities in the southeastern U.S. and throughout much of the world. They are all threatened to varying degrees and in different ways by rising seas and increased storm-induced flooding. Leaders in most of the affected states, counties, and localities appreciate these threats brought about by

climate change and are preparing for the future as best they can with limited knowledge and financial resources. Some of the plans are more scientifically sound than others. Some of the stakeholders are more receptive to change and to the investments of tax dollars than others. Recent impacts by Hurricane Ida on Louisiana and the Northeast may have increased the number of climate change believers. Most citizens and policy makers in the LECZ realize that "the water will come" [115] but are not sure what to do about it. Strengthening connections between scientists and state and federal agencies will certainly reduce uncertainties and mistrust of science.

Some of the past problems stem from ineffective communication and the fact that scientists have failed to convince non-scientists that the future holds some hitherto unexperienced, potentially devastating phenomena. Overly technocratic "lectures" or "town hall" meetings led by scientists tend to bore, and rarely convince, the public. Another part of the problem is that these events usually focus on information delivery, so scientists have "done all of the talking" in an effort to "raise awareness." Rarely do scientists and policy makers take time to listen and truly engage with stakeholders to understand their perspectives, morays, concerns, and experiences. Effective communication requires trust and a willingness to listen, which cannot manifest in events designed to "educate the public." This is changing and it must. One of the most urgent recommendations that emerges from the foregoing literature review is that serious engagement of stakeholders in substantive dialogue must precede adaptive planning. The formation of regional stakeholder advisory boards, the conduct of regular meetings and workshops, and the development of accessible web sites serving actionable information, interactive cyber toolkits, and information networks can help to sustain the necessary engagement. This is already happening in many counties and cities and those successes should be widely shared. Rigorous transdisciplinary science is essential to the future of LECZ communities in the southeastern US and globally, but collaborations must extend well beyond the halls of science.

To close, we must note that this review is not comprehensive and does not consider the threats and risks of all coastal environments and communities. It deals only with the processes and resilience of LECZ communities in the southeastern coastal states of the USA. These realms are not threatened by severe tectonic activity, earthquakes, tsunamis, or landslides (or by aggressive military forces). Although many of the SE US LECZ communities are impoverished and under-represented, their plight pales in comparison with those of the nations surrounding the Bay of Bengal, especially Bangladesh, as well as other Southeast Asian regions. Eventually, and hopefully soon, coastal vulnerabilities must be addressed from a global perspective, but we have not offered that here although others have begun to do so. We have tried to contribute to a start with a geographically limited review. We hope that this is a building block for future extensions, but we acknowledge that we have only offered a microcosmic assessment of a cosmic issue.

**Author Contributions:** All authors contributed ideas, perspectives, references, and text to this review. The corresponding author (LDW) also contributed original text and edited and synthesized input from all authors. All authors have read and agreed to the published version of the manuscript.

**Funding:** Preparation of this review was not supported by external grants or contracts.

**Acknowledgments:** This review resulted from a coastal resiliency initiative promoted and coordinated by the Coastal and Environmental Research Committee of the Southeastern Universities Research Association (SURA), in Washington, DC and followed two SURA-led workshops hosted by the University of South Florida in Tampa and St. Petersburg, FL.

**Conflicts of Interest:** The authors declare no conflict of interest.

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
