# Peer review of "Anticipating and Adapting to the Future Impacts of Climate Change on the Health, Security and Welfare of Low Elevation Coastal Zone (LECZ) Communities in Southeastern USA"

_jmse, doi:10.3390/jmse9111196_

Round 1

Reviewer 1 Report

The subject of the article is interesting and it is linked to the objectives of the journal, however, there are a number of issues that have to be reconsidered.

Q1: What period is referred to by recent literature? What sources were used? Any methodology? Even though it is not a systematic review, please provide a scheme that explains the flow of your research/ investigation. Illustrating the fundamental material presented in the extensive article could help to facilitate understanding of its internal structure.

Q2: references should be mentioned along with quotation marks, including page numbers.

Q3: Figures should be accompanied by the source. Short titles are preferable, additional explanation can be moved in the text

Q4: What were the criteria to choose the 4 study cases?

Q5: In section 7, how were the knowledge gaps identified?

Q6: Please provide more information on the implementation, potential barriers and risks related to the concept of network integration.

Q7: What are the limitation of the study?

Q8: Reference list is not in accordance with author's instructions

Author Response

jmse-1413381; Responses to Comments and Questions from Reviewer 1

Q1: What period is referred to by recent literature? What sources were used? Any methodology? Even though it is not a systematic review, please provide a scheme that explains the flow of your research/ investigation. Illustrating the fundamental material presented in the extensive article could help to facilitate understanding of its internal structure.

Response: All aspects of this question are now addressed. We have added a two-paragraph section 2 on Research Approach. Lines 139-186

Q2: references should be mentioned along with quotation marks, including page numbers.

Response: I am not quite sure what is being requested here but I assume that it refers specifically to the few cases where we have included direct quotes. I have gone through it and inserted the appropriate page numbers from which the quotes came.

Q3: Figures should be accompanied by the source. Short titles are preferable, additional explanation can be moved in the text

Response: Source details have been added to the Figure captions.

Q4: What were the criteria to choose the 4 study cases?

Response: This question is now answered by additional text in lines 595-598.

Q5: In section 7, how were the knowledge gaps identified?

Response: This question is now answered by additional text in lines 842-853

Q6: Please provide more information on the implementation, potential barriers and risks related to the concept of network integration.

Response: This question is now answered by additional text in lines 981-995.

Q7: What are the limitations of the study?

Response: This question is now answered by the new closing paragraph in lines 1062-1072.

Q8: Reference list is not in accordance with author's instructions

Response: On reviewing this, it appears to me that the references conform closely to the instructions to authors, so I am a bit confused by this. Based on recent experience with JMSE, I am hoping that any irregularities will be picked up and corrected by automated copy editing.

Reviewer 2 Report

The Authors present a review article about Low Elevation Coastal Zones and the several threats thats these zones face. Special attention is given to the impacts of Climate Change, and especially sea level rise and storm surges.

Four areas of low elevation are presented as case studies, and guidelines are given for the appropriate management of these zones, in order to mitigate risks and hazards.

I would like to advise for a careful review for some minor grammar-spelling errors. Other that that, the paper is generally very well written and I think it should be accepted for publication.

Author Response

jmse-1413381; Responses to Comments and Questions from Reviewer 2

Comments from Reviewer 2: I would like to advise for a careful review for some minor grammar-spelling errors. 

Response: We have carefully gone through the entire paper with MS Word spell-check and grammar check and corrected several errors.  We have also added text in several places in the revision to address comments and concerns from Reviewer 1.

Round 2

Reviewer 1 Report

The manuscript was updated with the recommendations. Regarding the last point about the references, they are not edited according to the instructions for authors (https://www.mdpi.com/journal/jmse/instructions#references). However, it remains editor's decision whether to ask for improvement.